# Superficial and Deep Capillary Plexuses: Potential Biomarkers of Focal Retinal Defects in Eyes Affected by Macular Idiopatic Epiretinal Membranes? A Pilot Study

**DOI:** 10.3390/diagnostics12123205

**Published:** 2022-12-17

**Authors:** Andrea Maria Coppe, Giuliana Lapucci, Luca Buzzonetti, Guido Ripandelli, Giancarlo Iarossi

**Affiliations:** 1Bambino Gesù Children’s Hospital—IRCCS, 00165 Rome, Italy; 2Studio Oculistico Coppé, 00199 Rome, Italy; 3G. B. Bietti Foundation—IRCCS, 00198 Rome, Italy

**Keywords:** optical coherence tomography angiography, microperimetry, non-invasive retinal imaging, biomarkers, idiopathic epiretinal membrane

## Abstract

Inner macular layers are the most involved in the retinal distortion caused by idiopathic epiretinal membrane (iERM). They represent the anatomical structures in which the superficial (SCP) and deep (DCP) capillary plexus are embedded. We quantified flow signal (FS) in these capillary plexuses using Swept Source OCT angiography to identify possible markers for postoperative outcome. The software ImageJ was used to quantify the FS in a 150 µm radius area around each point analyzed by MAIA microperimeter. In 16 patients with unilateral iERM, focal light sensitivity (FLS) in the para- and perimacular areas was measured to evaluate macular function in 24 points overlapping macular plexuses and compared with normal fellow eyes (FEs). *t*-Test for independent samples iERM eyes (iERMEs) vs. fellow eyes (FEs) and Pearson correlation coefficient of FS vs. FLS in each point were calculated. A level of *p* < 0.05 was accepted as statistically significant. As a whole, FLS was significantly higher in FEs vs. ERMEs (*p* < 0.001); FS in both SCP and DCP was not significantly different between ERMEs and FEs (*p* = 0.827, *p* = 0.791). Correlation in focal retinal areas between FLS and FS in ERMEs was significant in SCP (*p* = 0.002) and not significant in DCP (*p* = 0.205); in FEs was significant in both SCP (*p* < 0.001) and DCP (*p* = 0.022). As previously described, these defective areas were located mainly in sites of distortion of retinal layers; therefore, it can be hypothesized that a focal change in FS, occurring mostly in SCP, could be involved in the onset of the functional defect.

## 1. Introduction

Idiopathic epiretinal membranes (iERM) are a frequent macular pathology with a prevalence of 7–11.8% of the population. Epiretinal membranes are made of glial cells, retinal pigment epithelial (RPE) cells, macrophages, fibrocytes and collagen in varying proportions. IERM can stretch the underlying retina, depending on the cell type present in the ERM [1,2], causing different degree of distortion of macular superficial (SCP) and deep (DCP) capillary plexuses.

Patients with ERMs usually show visual symptoms as the conditions worsen, including reduced visual acuity and metamorphopsia. A thickening of inner nuclear layer (INL) has been reported more in patients with metamorphopsia than in those without [3]. A further study showed a relationship between metamorphopsia score and the maximum depth of the retinal fold suggesting a correlation between the severity of retinal folds caused by contracted iERM and visual function [4].

Previous works identified preoperative markers correlated with postoperative visual function in order to predict visual recovery after surgical treatment [5]. Recent studies with SD-OCT demonstrated the important role of the preoperative integrity of inner retinal layers to determine the visual function in patients affected by iERM [6,7].

Inner retinal layers are the anatomical structures in which the superficial (SCP) and deep (DCP) capillary plexus of the macula are located; therefore, we quantified flow signal (FS) in these retinal capillary plexuses using both Swept Source-OCT (SS-OCT) and Swept Source SS-OCT angiography (OCTA) to assess the preoperative anatomical status and identify possible markers for postoperative outcome in patients affected by iERM.

The aim of this study is, therefore, to quantify by means of OCTA the FS in the SCP and DCP in eyes affected by iERM and evaluate a possible correlation with FLS.

## 2. Materials and Methods

The study was conducted in a consecutive series of patients with unilateral idiopathic ERMs; 31 eyes of 16 patients, 16 eyes with ERM (ERMEs) and 15 normative fellow eyes (FEs), 9 women and 7 men, mean age 63.21 ± 5.34 years, were studied.

All procedures in this study adhered to the tenets of the Declaration of Helsinki and were approved by the investigational review board of Central Ethic Committee IRCCS Lazio. All subjects gave their informed consent after the aim of the study had been fully explained.

Before imaging, all patients underwent ophthalmic examination, including best-corrected visual acuity and fundus examination, performed by a retina specialist using a +90 diopters (D) lens. Inclusion criteria were the presence of iERM in one eye and normal morphology without any retinal changes in the fellow eye. Exclusion criteria were the presence of any retinal or choroidal disease such as retinal detachment, retino-vascular disease, AMD, diabetic retinopathy, glaucoma or ocular hypertension, a history of previous ocular laser or surgery, eyes with refractive errors > 3D, media opacities that prevented good visualization of the fundus, any associated systematic disorders (e.g., systemic corticosteroids intake, diabetes or hypertension) or vascular diseases without retinopathy, which would affect the FS.

The macula was assessed using the split-spectrum amplitude-decorrelation angiography with PLEX^®^ Elite 9000 (Version 1.5.0.15909; Carl Zeiss Meditec Inc., Dublin, CA, USA). It is a swept-source OCT (SS-OCT) and OCT angiography (SS-OCTA) that provides automated segmented enface OCT with analysis of different plexus: Superficial Capillary Plexus (SCP), Deep Capillary Plexus (DCP), Avascular Retina, Choriocapillaris (CC) and Choroid (Ch). OCTA on the PLEX^®^ Elite 9000 is generated with the OMAG^®^ algorithm (optical microangiography) [8,9], which utilizes the complete complex OCT data signal, including both amplitude and phase, to detect motion of red blood cells within sequential OCT B-scans performed repeatedly at the same location [9,10,11]. The device performed each acquisition at a speed of 100,000 A-scans per second, using as optical source a Swept Source tunable laser, center wavelength between 1040 and 1060 nm; axial resolution 6.3 µm, transverse resolution 20 µm. A cube scan was performed, 500 A-scan made up a B-scan, 500 horizontal B-scan were sampled in the scanning area to form a 6 mm × 6 mm three-dimensional data cube. The software, using an algorithm that is a prototype provided by the manufacturer (ARI Network—Zeiss), computed the analysis of different retinal layers, produced images of vessels merged and calculate the FS, as percentage using a 6 mm diameter ETDRS grid, in the different layers of the neuroepithelium in the 6 mm × 6 mm central area centered on the fixation point. The software Macular Density v0.6.1 (beta release, Zeiss Algorithm Development, ARI Network), produced binarized images of the 2 plexuses, as shown in Figure 1 and Figure 2, in which is also overlapped the MAIA grid and displayed a three-dimensional cube of each focal area.

The PLEX^®^ Elite 9000 analyzes retinal layers using infrared (IR) light produced by a swept source laser. This IR laser ray penetrates in deeper retinal layers with less bias than light used in Spectral Domain OCT. The instrument further reduces the segmentation errors related to the projection artifacts on DCP using a software, that subtracts the artifacts related to the SCP flow signal from the enface image of the DCP.

The software ImageJ (ver. 1.50e, NIH, USA), was used to quantify the FS in the focal area around each point analyzed by MAIA microperimeter (Centervue S.p.A., Padova, Italy). The MAIA microperimeter checks the fixation stability and gives an output of 3 possible fixation stability: stable, relatively stable and unstable; we repeated the microperimetry if necessary to analyze only the examinations with stable fixation and to have valid sensitivity values in each patient. To evaluate macular functional status, we measured the focal light sensitivity (FLS) in the para- and perimacular areas, as reported in previous papers [12,13,14], by means of MAIA microperimetry using focal stimulus.

We decided to assess the visual function outside the central fovea, assessed in previous studies [15,16], as the central fovea corresponds to an avascular zone (FAZ), in which there is no FS. Fovea can have high light sensitivity, even if it is an avascular zone (FAZ), because in this area all the retinal layers are fed by the choroid. In the extrafoveal area, inner retinal layers are fed only by the capillary plexuses, that are involved in the damages caused by the ERM contraction.

An ImageJ circle with a radius of 150 micron was used to select the focal area around each one of the 24 points analyzed by microperimeter MAIA. Thereafter, in each binarized focal area, we calculated the density of white pixels, representing the focal FS, and density was then correlated with the FLS at each point. As the aim of the study is to evaluate the correlation between morphological and functional parameters in each focal areas of the retina, the N of focal areas examined in every one of the 16 patients are 24; therefore, 24 × 16 = 384 for each capillary plexus were analyzed.

The eccentricity of the 24 points analyzed in each eye was between 2.5° and 5°, the 1° grid was not included for its location on the border of the FAZ. The 12 points with eccentricity 2.5° represent inner grid, 12 points at 5° outer grid (Figure 3).

Images with visible eye motion or blinking artefacts and with poor image quality were excluded (defined as signal strength lower than 7/10). OCTs were visually assessed by the same retina specialist (A.M.C.) to ensure proper segmentation of SCP and DCP.

By measuring macular capillary network with OCTA, an association was found between the presence of epiretinal membranes in pediatric age and the reduction in FS at the level SCP and DCP [17]. In order to evaluate whether a reduction in FS in the macular plexuses increases the risk of functional defects due to epiretinal membranes in children, we decided to set a preliminary study on adult population with better collaborative skills to obtain a more accurate measurement of flows in the SCP and DCP

The *t*-Test for independent samples iERMEs vs. FEs and the Pearson correlation coefficient (R^2^) of FS (density) vs. FSL (dB) in each point was calculated, comparing the same focal areas of the retina according to the grid between iERMEs and FEs. A level of *p* < 0.05 was accepted as statistically significant. The statistical analyses were performed using SPSS software version 15.0 (SPSS, Inc., Chicago, IL, USA).

## 3. Results

FS and FLS values of each 24 focal areas from anyone of the 16 patients were evaluated and compared as a whole between affected and fellow eyes; in iERMEs, mean FLS was 24.39 ± 4.19 dB, in FEs 25.57 ± 4.18 dB (*p* < 0.001), with a statistically significant difference, as displayed in Figure 1.

Mean difference in FS was not significant in both SCP and DCP (ERMEs SCP 111.00 ± 21.98, DCP 71.51 ± 16.52; FEs SCP 111.33 ± 17.08, DCP 71.80 ± 10.85—SCP *p* = 0.827, DCP *p* = 0.791) (Figure 2).

A significant mean reduction in light sensitivity was present in affected eyes (Table 1).

Interesting findings were standard deviations (SD) differences, for both SCP and DCP, between ERMEs and FEs, probably associated with focal changes in the retinal structure related to ERM contraction. In iERMEs, the correlation between FLS and FS was significant in SCP (*p* = 0.002) and not significant in DCP (*p* = 0.205); in FEs, was significant in both SCP (*p* < 0.001) and DCP (*p* = 0.022). The correlation in iERMEs is negative in both SCP (R^2^ = −0.164) and DCP (R^2^ = −0.69), whereas in FEs, it is positive in SCP (R^2^ = 0.279) and negative in DCP (R^2^ = −0.121), as reported in Table 2 and Figure 3.

Differences are present for FLS and SCP FS between inner and outer grids: inner vs. outer grid in FEs FLS *p* = 0.010; FS SCP *p* < 0.001; FS DCP *p* = 0.029; in iERMEs, FLS *p* = 0.017; FS SCP *p* < 0.001; FS DCP *p* = 0.723. Therefore, a further analysis was performed to check differences between the inner and the outer grid between affected (iERMEs) and fellow eyes (FEs).

Comparison between affected and fellow eyes:

Inner grid: in iERMEs, mean FLS was 24.94 ± 3.71 dB; in FEs, 26.12 ± 3.95 dB, statistically significant between fellow and affected eyes (*p* < 0.004). Mean FS in iERMEs was 106.57 ± 21.91 for SCP and 71.85 ± 16.34 for DCP; in FEs, 107.41 ± 15.03 for SCP and 70.55 ± 11.01 for DCP without reaching a statistically significant difference between the two groups (SCP *p* = 0.676, DCP *p* = 0.398). Correlation between FLS and FS is significant in SCP in both iERMEs (*p* = 0.047) and FEs (*p* < 0.001), and not significant in DCP, iERMEs (*p* = 0.182) and FEs (*p* = 0.195). Correlation is positive only in SCP of FEs (R^2^ = 0.371) and negative in DCP of FEs (R^2^ = −0.097), SCP (R^2^ = −0.153) and DCP (R^2^ = −0.103) of iERMEs.

Outer grid: in iERMEs, mean FLS was 23.85 ± 4.56 dB, in FEs, 25.03 ± 4.34 dB (*p* = 0.006), with a statistically significant difference between fellow and affected eyes. Mean FS in iERMEs were 115.43 ± 21.21 for SCP and 71.21 ± 16.74 for DCP, in FEs, 115.24 ± 18.12 for SCP and 73.05 ± 10.58 for DCP without a statistically significant difference (SCP *p* = 0.737, DCP *p* = 0.251). Correlation between FLS and FS is statistically significant in FEs only in SCP (*p* = 0.0001) and not significant in DCP (*p* = 0.115), whereas it is not significant in both capillary plexuses of iERMEs: SCP (*p* = 0.081) and DCP (*p* = 0.535). Correlation is positive only in SCP of FEs (R^2^ = 0.283) and negative in DCP of FEs (R^2^ = −0.118), SCP (R^2^ = −0.135) and DCP (R^2^ = −0.048) of iERMEs. Based on these data, results showed that, except for the change in the means values of FLS and FS, there is no significant difference between fellow and affected eyes related to the eccentricity of focal retinal areas (Table 2).

More altered focal areas (FS> or <[mean ± 2SD]) of affected eyes were analyzed to evaluate if a higher correlation between FLS and FS was present.

In these areas of iERMEs, the correlation between FLS and FS in SCP (N = 63) *p* = 0.003 R^2^ = −0.370 and in DCP (N = 45) *p* = 0.006 R^2^ = −0.402 were statistically significant. In these focal areas, therefore, negative correlation is significant in both capillary plexuses (Table 3).

## 4. Discussion

The data of this study are consistent with the hypothesis that FS could be a marker to assess the anatomical status and predict the functional status of focal retinal areas in eyes affected by iERM. ERM contraction causes focal functional defects, as shown by the mean FLS which is statistically lower in affected vs. fellow eyes. On the contrary, the mean FS were not significantly different between the two groups in both SCP and DCP; however, this apparent discrepancy can be explained by the fact that in eyes with ERM, the irregular contraction of the ERM causes focal displacement of blood flow, thus not affecting the mean values, mainly in SCP, and in DCP. Additionally, retinal blood flow in SCP and light sensitivity are significantly different in relation to eccentricity of the focal retinal areas.

Analysis of the correlation between morphological and functional parameters shows that focal functional alterations in the inner retina can occur as a consequence of focal structural change. The correlation between FLS and FS is significant in SCP in both affected and fellow eyes. These data validate a relation between the blood flow density and the functional status in focal retinal areas. Focal blood flow could be related to the conformation of the retinal structure, which can be modified by inner retinal layer shrinkage due to the entity of ERM contraction. An increase in blood flow, mainly in the SCP, is present in focal areas in which the structure of inner retina, such as in eyes affected by ERM, is more altered. We found a significant and negative correlation between FS and FLS in SCP of iERMs, whereas in FEs, the correlation between these parameters is significant and positive, showing, therefore, a different relation between FS and FLS in healthy and affected areas of the retina, as shown in Figure 4, in which appears an increase in SCP FS in the retinal area distorted by the iERM. The positive correlation between FS and FLS in FEs is consistent with the metabolism of the inner retina, in which when the photoreceptors are stimulated by light there is a positive correlation with the oxygen concentration [18]. On the other hand, a different effect on the metabolism of the inner retina could be caused by the retina distortion in the iERMEs, in which the increase in FS, even if associated to a higher oxygen concentration, shows a negative correlation with FLS. It is possible that in these cases, the increase in FS may reflect mainly the distortion of the inner retina playing, consequently, a more relevant role than the oxygen concentration.

A further validation that mechanical distortion may have an influence on blood flow density in focal retinal areas can be represented by the higher correlation between FLS and FS in the focal areas in which the blood flow density is more altered. R^2^ is higher when the correlation is calculated in groups of focal areas in which the values of FS differ more (±2SD) from the mean (Table 3). The increase in blood flow is higher in SCP, because this plexus is closer and more deformed by ERM. FS is significantly correlated with FLS also in DCP, if the statistical analysis is performed on the more altered focal areas (mean ± 2SD group), in which structural changes occur also in the deeper retinal layers, contiguous to the superficial.

It was previously reported that a thickening of INL was more present in patients with metamorphopsia than in those without, suggesting a correlation between metamorphopsia score and the maximum depth of the retinal fold [4,19] and supporting the hypothesis that in the ERM contraction the inner macular layers (where SCP and DCP merge) are mainly affected [6]. Our data, showing the presence of defects in the SCP and DCP related to the grade of contraction, are in accordance with other previous studies [6,20,21]. A further relevant finding is the correlation between morpho-functional parameters. The onset of metamorphopsia could be due to the distortion of the inner retinal layers, as it was assessed that a displacement of the Mueller cells in the inner retina [22] and not of the photoreceptors in the outer retina may be the cause of metamorphopsia. This is correlated with the possible function of Mueller cells as “optical fibers” transmitting images focused on the inner retina through the neuroepithelium to the photoreceptors layer in the outer retina, and it is consistent with our finding showing changes in the inner retinal layers. This hypothesis has also led us to choose FLS as a functional test, as it can represent a more accurate and objective tool to evaluate the retinal function [23,24], without being affected by just the displacement of the inner retinal layers as occurs in metamorphopsia. Moreover, in a previous electrophysiological study on patients affected by iERM, we showed that the functional defects mainly involve the inner retinal layers [25].

The presence of a more significant change in FS in the SCP could suggest that the retinal displacement due to the ERM tangential traction primary involves the inner retinal layers fed by the SCP, histologically corresponding mainly to ganglion cells (GC) layer, and subsequently affecting bipolar and amacrine cells and both inner and outer plexiform layers, fed by DCP. The GC layer is more injured by the mechanical damage and a previous study of ours reported that the GC layer can also be affected by an alteration occurring before the mechanical damage related to the contraction on the iERM [26]. Moreover, the functional damage is mostly correlated with the focal change in FS; this is consistent with our unpublished data, where we observed that FLS in patients with iERM was not correlated to the thickening of the macula measured with SD-OCT. Therefore, the thickening should not be a marker to decide if the patient should undergo pars plana vitrectomy (PPV) and peeling of the iERM.

The correlation between FS and FLS is more significant in patients with a higher difference from mean value of FS; therefore, it can be hypothesized that the severity of the functional retinal cell damage is correlated to the amount of FS change (Table 3).

Furthermore, the FS change and the statistically significant correlation found between SCP and FLS in affected eyes, support the hypothesis that FS focal alterations could be helpful to assess the anatomical integrity and the corresponding functional status of inner retinal layers. Consequently, FS and FLS parameters could be considered predictive markers to evaluate when patients should undergo surgical treatment and to predict post-operative outcome.

It has been previously reported that the segmentation of SCP and DCP [27] in patients with macular disorders varies among the current OCT instruments and could be inaccurate; nevertheless, evaluation of macular FS in SCP and DCP assessed with PLEX^®^ Elite 9000, infrared light SS-OCT, is more accurate with respect to SD-OCT. Additionally, the more significant capillary plexus in the affected eyes is the superficial, in which lower projection artifacts are involved, related just to the presence of epiretinal (clear) tissue. Even if the presence of projection artefacts can bias the evaluation of the FS in the DCP, in the PLEX^®^ Elite 9000, the new software added to the infrared light of the SS-OCT significantly reduces the interference in the evaluation of the DCP. A limitation of the study is related to the series with N = 16; however, it should be considered that the number of focal retinal areas is 24 in each patient; therefore, as previously reported, 384 is the total number of focal areas examined in the study. Each focal area is an example of what happens in the retinal tissues as a consequence of ERM contraction. Following this model, we considered the N of 384 focal areas could be statistically significant to support a hypothesis reported in this pilot study. On the basis of these data, a further study in larger series is recommended to confirm that SCP could represent an interesting biomarker to assess focal retinal defects in eyes affected by iERM.

## 5. Conclusions

The evaluation of FS with new generation SS-OCTA, could represent a reliable technique to analyze the macular function in eyes affected by iERM, as demonstrated by the findings of correlation between FLS and FS in macular capillary network. These results allowed us to suggest that FS, mainly in SCP, can be a biomarker to detect morpho-functional worsening in patients affected by iERM and to decide when it is the optimal time, with best benefit-risk ratio, to perform pars plana vitrectomy and peeling of the macular epiretinal membrane.

## Data Availability

Not applicable.

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
