# Peer review of "Superficial and Deep Capillary Plexuses: Potential Biomarkers of Focal Retinal Defects in Eyes Affected by Macular Idiopatic Epiretinal Membranes? A Pilot Study"

_diagnostics, 2022, doi:10.3390/diagnostics12123205_

Round 1

Reviewer 1 Report

The Authors have assessed the superficial and deep capillary plexuses as potential  biomarkers of focal retinal defects in eyes affected by macular idiopathic epiretinal membranes.

The paper is generally well written.  The manuscript is correct in terms of the structure and the merits. The material and the methods are clearly described, the statistical analysis is performed appropriately.  

Major remarks and concerns:

The main disadvantage of this manuscript is a small sample of study patients and it should be highlighted as a limitation of the study. The conclusion drawn by the Authors that FS can be a "biomarker to detect morpho-functional worsening in patients affected by iERM and to decide when it is the optimal time, with best benefit-risk ratio, to perform pars plana vitrectomy and peeling of the macular epiretinal membrane” is strongly overassessed based only on 16 examined eyes. 

Minor remarks: abstract line 5 change 150 mm for 150 micron

Author Response

Reviewer's comment: Major remarks and concerns:

The main disadvantage of this manuscript is a small sample of study patients and it should be highlighted as a limitation of the study. The conclusion drawn by the Authors that FS can be a "biomarker to detect morpho-functional worsening in patients affected by iERM and to decide when it is the optimal time, with best benefit-risk ratio, to perform pars plana vitrectomy and peeling of the macular epiretinal membrane” is strongly overassessed based only on 16 examined eyes. 

Response: We thank the reviewer for the suggestion. We modified the title: “Superficial and Deep Capillary Plexuses: Potential Biomarkers of Focal Retinal Defects in Eyes Affected by Macular Idiopatic Epiretinal Membranes? A Pilot Study” (line 4) and we added this phrase in the Discussion, lines 310-318:

“A limitation of the study is related to the relatively small number of patients (N=16); however, it should be considered that the number of focal retinal areas is 24 in each patient; therefore, the total of focal areas examined in the study results in 384. Each focal area is representative of what happens in the retinal tissues as a consequence of ERM contraction. Following this model, we considered the N of 384 focal areas as statistically significant to support a hypothesis in a pilot study. On the basis of these data, a further study on a larger series is recommended to confirm that SCP could represent an interesting biomarker to assess focal retinal defects in eyes affected by iERM”

Reviewer's comment: Minor remarks: abstract line 5 change 150 mm for 150 micron

Response: Thank you for your notice, we corrected the mistake and we modified the text “mm” with “µm” in Abstract, line 14

Reviewer 2 Report

Authors have well written the manuscript titled “Superficial and Deep Capillary Plexuses: Potential Biomarkers of Focal Retinal Defects in Eyes Affected by Macular Idiopathic Epiretinal Membranes?”

I have few comments regarding this study:

·         Its not clear why authors conducted this study as microperimetry is a time taking clinical procedure and authors have enrolled only 16 subjects with iERM and mean age is also >65 years.

·         In general, retinal sensitivity is reduced in this age groups.

·         And authors have not showed how valid the sensitivity values as this age group people have more fixation issues and fatigue during test.

·         FLS is not clinically meaningful different between iERM and FE groups

·         Introduction: page 2; lines 52-66, authors have added lot of methodological explanation here. Please move to relevant locations in methods or discussion.

·         Methods:  Page 4, line 122, why authors have chosen 150 microns around the FLS loci? Any rationale?

·         Authors have not really explain/discuss why positive correlations in FE and negative correlations in iERM’s?

·         And these correlations are very poor for all

·         Discussion: Page 10, Line 218-220, how functional change can lead to structural alterations in the retina (usually its vice versa)?

Author Response

Reviewer's comment:  Authors have well written the manuscript titled “Superficial and Deep Capillary Plexuses: Potential Biomarkers of Focal Retinal Defects in Eyes Affected by Macular Idiopathic Epiretinal Membranes?”

I have few comments regarding this study:

Its not clear why authors conducted this study as microperimetry is a time taking clinical procedure and authors have enrolled only 16 subjects with iERM and mean age is also >65 years.

In general, retinal sensitivity is reduced in this age groups.

And authors have not showed how valid the sensitivity values as this age group people have more fixation issues and fatigue during test.

Response:  We performed microperimetry because allows the evaluation of the focal functional defects typical of patients with iERM. The whole time requested to perform this examination is approximately 20’ and results are quickly analysed; therefore, we thought that could be a useful technique to decide when to perform surgery to avoid further increases of the structural and functional retinal damages.

The mean age of the 16 patients was 63,21±5.34 yrs, only 6 patients were older than 65 years. The MAIA microperimeter checks the fixation stability and gives an output of 3 possible fixation stability: stable, relatively stable and unstable; we repeated the microperimetry, if necessary to analyse only the examinations with stable fixation and have valid sensitivity values in each patient. We added this previous phrase in Methods, lines 107-110

Reviewer's comment: FLS is not clinically meaningful different between iERM and FE groups

Response: FLS was significantly different between iERM and FE groups, it is reported in Results, line 148-150, and in Graph 1. To highlight this significant difference, we added the “p<0.001” on the image of the Graph 1

Reviewer's comment: Introduction: page 2; lines 52-66, authors have added lot of methodological explanation here. Please move to relevant locations in methods or discussion.

Response: Thank you for this comment, we moved these explanations in the Method paragraph (lines 110-118, 135-140)

Reviewer's comment: Methods:  Page 4, line 122, why authors have chosen 150 microns around the FLS loci? Any rationale?

Response: We chose 300-micron diameter, 1° degree, to have a compromise that allows sufficient surface to evaluate FS and to have enough distance among points, and from inner and outer grids

Reviewer's comment: Authors have not really explain/discuss why positive correlations in FE and negative correlations in iERM’s?

Response: We added the following phrase to discuss about these data in Discussion (lines 233-240): “The positive correlation between FS and FLS in FEs is consistent with the metabolism of the inner retina, in which when the photoreceptors are stimulated by light there is a positive correlation with the oxygen concentration. (Linsenmeier RA, Zhang HF. Retinal oxygen: from animals to humans. Prog Retin Eye Res. 2017;58:115-151. doi:10.1016/j.preteyeres.2017.01.003). On the other hand, a different effect on the metabolism of the inner retina could be caused by the retina distortion in the iERMEs, in which the increase of FS, even if associated to a higher oxygen concentration, shows a negative correlation with FLS. It is possible that in these cases the increase of FS may reflect mainly the distortion of the inner retina playing, consequently, a more relevant role than the oxygen concentration.”

Reviewer's comment: And these correlations are very poor for all

Response: The correlations for the SCP are statistically significant in both FEs and iERMEs, thus, showing a robust numerical strength

Reviewer's comment: Discussion: Page 10, Line 218-220, how functional change can lead to structural alterations in the retina (usually its vice versa)?

Response: Thank you for your observation. It was a mistake in digitation and we modified the phrase (lines 221-223) as follows: “Analysis of correlation between morphological and functional parameters shows that focal functional alterations in the inner retina can occur as a consequence of focal structural change”

Round 2

Reviewer 2 Report

No further comments